# Temporal Weighted Encoding:
# Towards Maximal-Capacity Spike Coding for ANN-SNN Conversion

Yiwen Gu [1]   Junchuan Gu [1]   Haibin Shen [1]   Kejie Huang [1]

## Abstract

Spiking Neural Networks (SNNs) emulate the spiking behavior of biological neurons and are promising for energy-efficient neuromorphic computing. A widely used strategy to train SNNs is to convert pretrained Artificial Neural Networks (ANNs), where the accuracy and efficiency are determined by the spike encoding scheme. Traditional methods based on spike count or timing severely underutilize the available encoding space, leading to large accuracy degradation under low-timestep constraints. More expressive alternatives involve complex dynamics, which hinder scalability and practical deployment. To address these challenges, we propose Temporal Weighted Encoding (TWE). Spikes are implicitly assigned exponentially decaying weights through a recursive integration, drawing an analogy to a temporal bit sequence. We systematically analyze the temporal mismatch caused by this weight pattern and propose temporal relaxation and threshold relaxation to resolve this issue, enabling fast and accurate activation encoding. Extensive experiments demonstrate that TWE achieves negligible conversion loss with significantly fewer timesteps, offering a scalable and efficient solution for SNN deployment.

## 1. Introduction

Spiking Neural Networks (SNNs), widely regarded as the third generation of neural networks, draw inspiration from the biological structure and functional dynamics of the brain (Maass, 1997). Unlike conventional Artificial Neural Networks (ANNs) that operate on continuous real-valued activations, SNNs transmit information through discrete spike

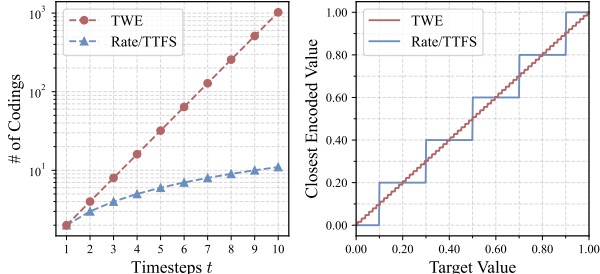

Figure 1. Comparison of encoding capacity and precision. Left: The number of codings available across different timesteps (log scale). TWE achieves an exponential growth of $2^T$, significantly outperforming the linear growth of rate/TTFS coding. Right: Encoding granularity for values in $[0, 1]$ with $t = 6$. TWE can approximate continuous values with high fidelity, whereas rate/TTFS coding suffers from coarse quantization steps.

events, enabling a natural representation of temporal dynamics and event-driven computation (Taherkhani et al., 2020). This aligns with the brain's energy-efficient processing paradigm, making SNNs a promising candidate for low-power and efficient neuromorphic systems (Yamazaki et al., 2022).

Currently, two dominant learning algorithms exist for SNNs: gradient-based optimization and ANN-SNN conversion. In gradient-based methods, the non-differentiable spike generation process is replaced by surrogate gradients or Spike-Timing-Dependent Plasticity (STDP) mechanisms (Neftci et al., 2019; Lee et al., 2016; Wu et al., 2018; Bellec et al., 2019). However, this inevitably introduces a mismatch between the training and inference dynamics, which limits performance. In addition, the training process still relies on conventional GPUs, which are not well suited to exploit the sparsity of spikes.

In contrast, ANN-SNN conversion treats spiking neurons as temporal encoders. Using the same network parameters, neuron dynamics are carefully designed to fire spikes that encodes the original ANN activations (Figure 2, left pannel). Owing to their stability and scalability, conversion-based

[1] College of Information Science & Electronic Engineering, Zhejiang University, Hangzhou, Zhejiang Province, China. Correspondence to: Kejie Huang <huangkejie@zju.edu.cn>.

*Proceedings of the 43rd International Conference on Machine Learning*, Seoul, South Korea. PMLR 306, 2026. Copyright 2026 by the author(s).

methods have produced many of the best-performing SNNs to date. (Hu et al., 2023; Hao et al., 2023; You et al., 2024; Huang et al., 2025a; Zhao et al., 2025).

In this framework, the choice of spike encoding schemes becomes the central bottleneck. Given $T$ timesteps, the presence or absence of a spike at each step spans a binary encoding space $\mathcal{V}_s = \{0, 1\}^T$. Theoretically, this allows for exponentially large information capacity. Despite this vast volumn, existing encoding schemes utilize only a small fraction of it.

Rate coding relies solely on spike count. This collapses a combinatorial number of spike sequences into a single value, resulting in huge encoding redundancy. Time-To-First-Spike (TTFS) coding, on the other hand, restricts valid codes to single-spike sequences, leaving the majority of $\mathcal{V}_s$ undefined. While more expressive temporal encodings have been explored (Park et al., 2019; Stöckl & Maass, 2021; Han & Roy, 2020; Huang et al., 2025a; Rueckauer & Liu, 2021), they typically rely on complex dynamics and may compromise the event-driven nature of SNNs. As a result, primitive rate coding remains the dominant paradigm in practice. This reveals a fundamental tension between encoding efficiency and implementational simplicity. How to achieve maximal utilization of the encoding space (i.e., achieve maximal capacity) under simple integrate-and-fire dynamics remains an open problem.

To address this, we revisit the structure of $\mathcal{V}_s = \{0, 1\}^T$. A key insight is that this encoding space is mathematically isomorphic to a digital bit stream of length $T$. In digital systems, binary representation provides a canonical example of a *bijective* mapping between codes and values, which ensures maximal utilization of the coding capacity. Motivated by this analogy, we treat spikes as a sequence of temporal bits and introduce Temporal Weighted Encoding (TWE). We show that through a simple recursive integration, spikes are *implicitly* assigned exponentially decaying weights, implementing a temporal counterpart of binary representation.

However, since ANN activations are typically dominated by small values, TWE induces a skewed spike distribution: spikes occur more frequently at later timesteps, while early spikes are relatively rare. As a result, the accumulated membrane current tends to lag in the temporal domain, which makes online encoding[1] challenging (Figure 3). One straightforward strategy is to delay the firing decisions so that the input current can be accumulated over a longer time horizon, which we term **temporal relaxation**. To reduce the required delay, we further propose a complementary mechanism termed **threshold relaxation**. The firing threshold is lowered to compensate for the temporal lag. In this set-

---

[1]We use "online encoding" to denote the simultaneous generation of spikes, where the output begins at the same timestep as the input arrives.

ting, negative spikes (Hu et al., 2023; Li et al., 2022; Wang et al., 2022; Guo et al., 2024) are introduced to prevent overshooting the target value.

We validate TWE with extensive experiments across various network architectures. The results demonstrate that the proposed method substantially reduces the number of timesteps while preserving the performance of the original ANN.

## 2. Related Works

Various spike coding schemes have been explored to support ANN-SNN conversion. Among them, rate coding remains the most prevalent paradigm in practice due to its simplicity of implementation. (Rueckauer et al., 2017; Hu et al., 2023; You et al., 2024) Rate coding represents information by the total number of spikes, disregarding their precise timing. Although this improves robustness, a large amount of representational capacity is wasted. Prior efforts have focused on mitigating conversion errors under rate coding, including pre-quantization (Li et al., 2022), activation clipping or shifting (Hao et al., 2023; Bu et al., 2022), and the use of negative spikes (Hu et al., 2023; Li et al., 2022). However, the fundamental limitation in terms of coding efficiency remains largely unaddressed.

To promote spike sparsity, alternative schemes such as Time-To-First-Spike (TTFS) coding have been adopted (Zhao et al., 2025; Yang et al., 2024a; Stanojevic et al., 2023). TTFS encodes information in the inverse of spike latency, where an earlier spike indicates a stronger input. While the single-spike constraint minimizes energy consumption, it severely restricts the encoding capacity. Consequently, TTFS typically requires thousands of timesteps to achieve sufficient precision. Moreover, TTFS operates in an offline manner: each layer can only start computing after receiving all inputs from the previous layer. This leads to substantial end-to-end latency and has not been effectively resolved in existing works.

Beyond these biologically plausible schemes, recent works have explored more expressive but complex dynamics. Burst coding methods (Park et al., 2019; Wang et al., 2025) allow neurons to emit multiple spikes within a single timestep, while multi-threshold neurons (Huang et al., 2025b;a) trigger spikes with different magnitudes. Phase coding (Rueckauer & Liu, 2021; Kim et al., 2018) assigns power-of-two weights to spikes, sharing the same weight structure as TWE. From an implementation perspective, these methods can be viewed as explicitly modulating spikes and thresholds with time-varying coefficients (Stöckl & Maass, 2021). This poses practical challenges for efficient neuromorphic implementation and may compromise the event-driven nature of SNNs.

In contrast, in TWE the spike weights arise from recur-

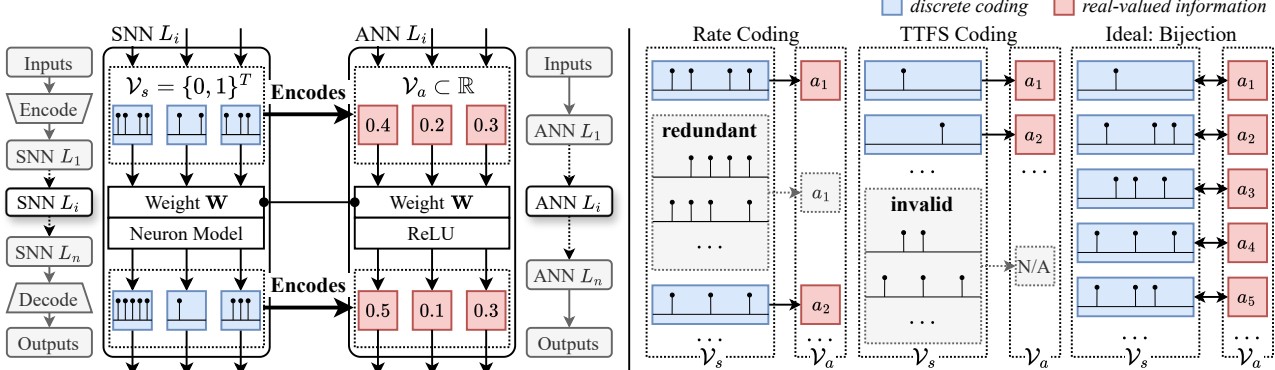

*Figure 2.* Overview of the ANN-SNN conversion framework and encoding analysis. The dashed boxes denote the spike encoding space $\mathcal{V}_s = \{0,1\}^T$ and the activation space $\mathcal{V}_a$. The blue and red shaded regions represent individual elements in $\mathcal{V}_s$ and $\mathcal{V}_a$, respectively. Left: The converted SNN inherits weights $\mathbf{W}$ from a pretrained ANN. The core objective is to map spike sequences in $\mathcal{V}_s$ to activations in $\mathcal{V}_a$ via spiking neuron dynamics. Right: Analysis of common encoding schemes. Rate coding suffers from redundancy, where multiple spike sequences correspond to the same activation value, while TTFS coding underutilizes the spike space by restricting valid codes. An ideal encoding establishes a bijective mapping between $\mathcal{V}_s$ and $\mathcal{V}_a$, thereby maximizing the encoding capacity.

sive integration. All spikes retain identical magnitudes and the threshold remains fixed, avoiding the complexity issue. More importantly, while previous works (Stöckl & Maass, 2021; Rueckauer & Liu, 2021) rely on offline encoding (similar to TTFS), we systematically analyze this problem and introduce a threshold relaxation mechanism to enable low-latency computation.

## 3. Preliminaries

### 3.1. Integrate-and-Fire Dynamics

Spiking neurons form the fundamental building blocks of SNNs. Each neuron integrates incoming spikes into its membrane potential and emits a spike when this potential exceeds a firing threshold $v_{th}$. A general discretized neuron model can be described as follows:

$$
\begin{cases}
o_i^l[t] = D(u_i^l[t-1], z_i^l[t]), \\
s_i^l[t] = H(o_i^l[t] - v_{th}^l), \\
u_i^l[t] = o_i^l[t] - \theta^l s[t].
\end{cases}
$$

Here, $i$ and $l$ denote the neuron and layer index, respectively. $o_i^l[t]$ and $u_i^l[t]$ represent the membrane potential before and after reset. The function $D(\cdot)$ defines the generalized membrane dynamics, which updates the potential based on previous state and current input $z_i^l[t]$. The output spike $s_i^l[t]$ is determined by the Heaviside step function $H(\cdot)$, which outputs 1 if the argument is non-negative and 0 otherwise. $\theta^l$ denotes the spike magnitude, which is typically set to $v_{th}^l$. It serves as a scaling factor that maps the boolean spike event to a scalar quantity.

The input current $z_i^l[t]$ aggregates the spikes from the preceding layer:

$$
z_i^l[t] = \theta^{l-1} \sum_j w_{ij}^l s_j^{l-1}[t] + b_i^l,
$$

where $w_{ij}^l$ and $b_i^l$ represent the synaptic weight and bias, respectively. $\theta^{l-1}$ is multiplied here to restore the magnitude of the transmitted spikes. Note that this scaling factor can be absorbed into the weight parameters to keep the network event-driven.

### 3.2. ANN-SNN Conversion

The core principle of ANN-SNN conversion is to encode ANN activations $a$ into discrete spike trains $s[t]$, as illustrated in the left panel of Figure 2. Specifically, the incoming spike trains $s^{l-1}[t]$, which encode the activations $a^{l-1}$, are first aggregated via synaptic weights into input currents. Neurons then decode these inputs and generate the output spike trains $s^l[t]$, which should accurately encode the target activations $a^l$. Although described here at the sequence level for clarity, the actual computation proceeds in a step-by-step manner over time. This requires precise alignment between the encoding scheme and the neuron dynamics.

## 4. Methods

From an information-theoretic perspective, a spike train of length $T$ provides an encoding space $\mathcal{V}_s = \{0,1\}^T$, which is structurally equivalent to a digital bit stream. This naturally motivates the treatment of spike encoding as a temporal bit-sequence representation. Following the canonical binary

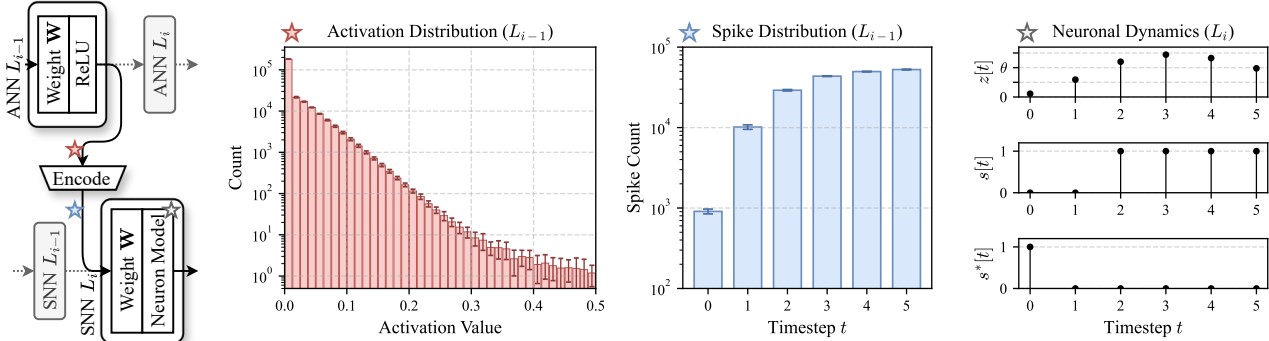

*Figure 3.* Illustration of the temporal mismatch in TWE. The leftmost schematic shows the experimental setup. To avoid cumulative encoding errors, ANN activations from layer $L_{i-1}$ are directly encoded to drive SNN layer $L_i$. The activation distribution in $L_{i-1}$ is dominated by small values (red ★). Consequently, the resulting spike distribution is heavily skewed toward later timesteps (blue ★), since small activations are represented by low-weight spikes. Error bars indicate the variation across different samples within a batch of 128 inputs. Grey ★: Dynamics of a representative neuron in $L_i$. The accumulated input current grows too slowly to support the early firing required by the target spike pattern $s^*[t]$. This temporal mismatch prevents the actual output $s[t]$ from fully encoding the input.

representation, we define **Temporal Weighted Encoding (TWE)** as:

$$R_{\text{TWE}}(s) \triangleq \theta \sum_{t=1}^{T} s[t]\, 2^{T-t}.$$

The spike magnitude $\theta$ is set to match the activation range during ANN-SNN conversion. TWE can uniformly encode $2^T$ values with $T$ timesteps, far exceeding the $T$ values achievable with conventional rate or TTFS coding, as illustrated in Figure 1.

### 4.1. Integration Dynamics

The choice of power-of-two weights enables a simple recursive decoding process. Specifically,

$$o_i^l[t] = D(u_i^l[t-1], z_i^l[t]) = \underbrace{2 \times u_i^l[t-1]}_{\text{recursive}} + z_i^l[t]. \quad (1)$$

Unrolling this recursion reveals that after $T$ timesteps, the accumulated potential exactly recovers the weighted sum defined by TWE.

The recursive integration dynamics can be interpreted as modelling the *relative* significance of spikes: The effective weight for the current timestep is always normalized to unity. This eliminates the need to temporally modulate thresholds or input currents (Stöckl & Maass, 2021; Rueckauer & Liu, 2021; Huang et al., 2025a), maintaining compatibility with event-driven implementations.

### 4.2. Firing Dynamics

Define the total weighted input $I(z) \triangleq \sum_t 2^{T-t} z[t]$. The encoding goal is to generate an ideal spike pattern $s^*[t]$,

such that $R_{\text{TWE}}(s^*) = I(z)$. Although TWE offers high encoding capacity with straightforward decoding, achieving accurate online encoding presents a fundamental challenge. We illustrate this issue in Figure 3 using an illustrative setup. Specifically, the output activations of ANN layer $L_{i-1}$ are encoded into TWE spike trains and fed to SNN layer $L$, where we examine the layer's encoding behavior.

**Revealing Temporal Mismatch.** As ANN activations are typically dominated by small values (Figure 3, red ★), the encoded TWE spike trains are heavily skewed towards later timesteps (Figure 3, blue ★). This causes the aggregated input $z[t]$ to lag in time and creates a causality dilemma: when $I(z)$ is large, a skewed $z[t]$ often fails to trigger early (high-weight) spikes required by $s^*[t]$. As a result, neurons often undershoot the target value, leading to encoding errors (Figure 3, grey ★). We refer to this phenomenon as **temporal mismatch**: the input current lags behind the ideal firing schedule in the temporal domain.

**Temporal Relaxation.** To bridge the gap between lagging inputs and the ideal output, a straightforward strategy is to delay the firing decisions, allowing for sufficient input accumulation. We introduce an output delay parameter $T_r$, and define the firing dynamics as:

$$s_i^l[t] = \mathbb{I}_{t \geq T_r} \cdot H(o_i^l[t] - 2^{T_r} v_{th}^l), \quad (2)$$

where $\mathbb{I}_{t \geq T_r}$ is an indicator function that enforces the delay, which can be implemented by a simple gate signal. The firing threshold is scaled by $2^{T_r}$ to compensate for the exponential amplification of membrane potential during the delay period.

This strategy is termed **temporal relaxation** as it relaxes

the temporal alignment constraint between inputs and ideal outputs.

**Definition 4.1** (Feasible input space)**.** Consider a bounded input space $\mathcal{V}_z = [-Z, Z]^T$. Given an ideal output $s^*[t]$, the **feasible input space** $\mathcal{Z}(T_r, s^*)$ is defined as the subset of inputs $z \in \mathcal{V}_z$ which trigger the target output:

$$\mathcal{Z}(T_r, s^*) = \{z \in \mathcal{V}_z \mid s[t + T_r] = s^*[t], \ \forall t\}.$$

**Theorem 4.2** (Temporal relaxation)**.** *Under the assumption of bounded inputs, for any target $s^*[t]$:*

1. *The volume of the feasible input space grows exponentially with the delay:*

$$\frac{\text{Vol}(\mathcal{Z}(T_r, s^*))}{\text{Vol}(\mathcal{Z}(0, s^*))} \approx 2^{T_r}.$$

2. *When $T_r = T$, the feasible space covers all valid inputs, ensuring ideal encoding: $s[t + T_r] \equiv s^*[t]$.*

The proof is provided in Section A. Theorem 4.2 suggests that temporal relaxation expands the feasible input space, allowing neurons to accommodate more input conditions. This improves both encoding accuracy and robustness. In the extreme case where $T_r = T$, the encoding degenerates into an offline scheme, which is used in several other schemes (e.g., TTFS coding (Zhao et al., 2025; Stanojevic et al., 2023) phase coding (Rueckauer & Liu, 2021) and few-spike coding (Stöckl & Maass, 2021)). Although accurate, such a setting incurs high latency. To achieve high-fidelity encoding with minimal delay, we further introduce a **threshold relaxation** mechanism.

**Threshold Relaxtion.**   Since temporal mismatch fundamentally arises from the lagging input, a natural remedy is to lower the firing threshold to encourage earlier responses. Specifically, we introduce a relaxation factor $\alpha \in (0, 1)$ and set the threshold as $v_{th}^l = \alpha \theta^l$.

This design relies on a mild assumption: early input evidence often indicates continued or stronger accumulation in subsequent timesteps. Under this assumption, lowering the threshold enables proactive encoding that anticipates future inputs, rather than causing systematic over-encoding. To fundamentally prevent the risk of overshooting, we further allow neurons to emit negative spikes for error correction. We note that signed spikes are well-established in recent SNN literature and pose no significant implementation overhead. The resulting firing dynamics are defined as:

$$s_i^l[t] = \mathbb{I}_{t \geq T_r} \cdot [H(o_i^l[t] - 2^{T_r} v_{th}^l) - \underbrace{H(-o_i^l[t] - 2^{T_r} v_{th}^l)}_{\text{negative spike trigger}}].$$

$$\text{(3)}$$

**Proposition 4.3** (Encoding error analysis)**.** *For any input $z \in [-Z, Z]^T$ and its ideal encoding $s^*[t]$, the encoding error is proportional to the residual membrane potential:*

$$R_{\text{TWE}}(s) - R_{\text{TWE}}(s^*) = -\frac{1}{2^{T_r}} u_i^l[T + T_r].$$

The proof is provided in Section A. Proposition Theorem 4.3 establishes a direct link between encoding accuracy and the residual membrane potential $u[t]$. To minimize the encoding error, we thus analyze the dynamics of $u[t]$.

**Theorem 4.4** (Threshold relaxation)**.** *Consider a bounded input $|z[t]| \leq Z$. Let $\beta = \max(\alpha, 1 - \alpha)$. If the spike magnitude $\theta \geq \frac{Z}{1-\beta}$, then the probability density function of $u[t]$, denoted by $f_t(u)$, satisfies:*

1. *$f_t(u)$ converges to a uniform invariant measure supported on $\Omega = [-\beta\theta, \beta\theta]$:*

$$\lim_{t \to \infty} f_t(u) = \mathcal{U}[-\beta\theta, \beta\theta] = \begin{cases} \frac{1}{2\beta\theta}, \text{if } |u| \leq \beta\theta, \\ 0, \text{otherwise}. \end{cases}$$

2. *The variance $Var(u) = \mathbb{E}[u^2]$ is minimized when $\alpha = 0.5$:*

$$\min_\alpha Var(u) = \frac{\theta^2}{12} \quad at \quad \alpha = 0.5.$$

The proof is provided in Section A.

*Remark* 4.5. Theorem 4.4 assumes that $\theta \geq \frac{Z}{1-\beta}$. This condition can always be satisfied by choosing a sufficiently large threshold, since the maximum input current is bounded by $Z = \sum_j |\omega_{ij}|$. However, an excessively large threshold trades off encoding precision, as the neuron becomes insensitive to small inputs. In our experiments, we set $\theta$ to match the activation range of the original ANN (see Section 5 for details), which may violate this condition in some cases. Nevertheless, we empirically find that the theoretical result remains robust, as $\alpha = 0.5$ provides a balanced tolerance for both positive and negative errors.

From another perspective, the introduction of negative spikes creates representation redundancy in the target sequence $s^*[t]$. For instance, a value encoded as $(0, 1)$ in standard binary can be equivalently represented as $(1, -1)$ using signed spikes (since $0 \cdot 2^1 + 1 \cdot 2^0 \equiv 1 \cdot 2^1 - 1 \cdot 2^0$). Consequently, this redundancy enlarges the feasible input space $\mathcal{Z}(T_r, s^*)$, as the input current is no longer constrained to trigger a unique spike pattern but can converge to any of the valid equivalent forms.

*Table 1.* Comparison of TWE with other encoding methods for ANN-SNN conversion on CIFAR-10, CIFAR-100, and ImageNet. We report ANN accuracy, SNN accuracy, accuracy difference ($\Delta$ Acc.), and the required encoding timesteps $T$. Results of TWE are highlighted in gray. As Ternary Spike (highlighted in cyan) is a direct training baseline, the coding scheme, ANN accuracy, and accuracy difference are denoted by "-". The best results are in bold and the second-best are underlined.

| | Method | Coding Scheme | Arch. | ANN Acc. | SNN Acc. | $\Delta$ Acc. ↑ | T ↓ |
|---|---|---|---|---|---|---|---|
| **CIFAR-10** | Ternary Spike[†] (Guo et al., 2024) | - | ResNet-19 | - | 95.80% | - | **2** |
| | FS-Conversion (Stöckl & Maass, 2021) | FS | ResNet-20 | 91.58% | 91.45% | -0.13% | 10 |
| | TSC (Han & Roy, 2020) | TSC | VGG-16 | 93.63% | 93.57% | -0.06% | 512 |
| | TTFS Mapping (Stanojevic et al., 2024) | TTFS | VGG-16 | 93.68% | 93.59% | +0.01% | 1024 |
| | Fast-SNN (Hu et al., 2023) | Rate | ResNet-18 | 95.62% | 95.57% | -0.05% | 7 |
| | QCFS (Bu et al., 2022) | Rate | ResNet-18 | 96.04% | 95.92% | -0.12 | 16 |
| | **TWE (ours)** | TWE | VGG-16 | 95.61% | 95.66% | **+0.05%** | 3 |
| | | | ResNet-18 | 96.33% | 96.34% | +0.01% | 3 |
| **CIFAR-100** | TSC (Han & Roy, 2020) | TSC | ResNet-20 | 68.72% | 67.81% | -0.91% | 1024 |
| | TTFS Mapping (Stanojevic et al., 2024) | TTFS | VGG-16 | 72.20% | 72.20% | **+0.00%** | 1024 |
| | COS (Hao et al., 2023) | Rate | VGG-16 | 76.35% | 76.26% | -0.09% | 4 |
| | QCFS (Bu et al., 2022) | Rate | VGG-16 | 76.34% | 76.24% | -0.10% | 16 |
| | **TWE (ours)** | TWE | VGG-16 | 77.21% | 77.20% | -0.01% | **3** |
| **ImageNet** | Ternary Spike[†] (Guo et al., 2024) | - | ResNet-34 | - | 70.74% | - | 4 |
| | FS-Conversion (Stöckl & Maass, 2021) | FS | ResNet-50 | 75.22% | 75.10% | -0.12% | 10 |
| | TSC (Han & Roy, 2020) | TSC | ResNet-34 | 70.64% | 69.93% | -0.71% | 4096 |
| | Fast-SNN (Hu et al., 2023) | Rate | VGG-16 | 73.02% | 72.95% | -0.07% | 7 |
| | QFFS (Li et al., 2022) | Rate | VGG-16 | 73.08% | 73.10% | +0.02% | 8 |
| | SpikeZIP-TF (You et al., 2024) | Rate | ViT-S/16 | 81.50% | 81.45% | -0.05% | 64 |
| | TTFSFormer (Zhao et al., 2025) | TTFS | ViT-S/16 | 81.40% | 81.38% | -0.02% | 4096 |
| | DCGS (Huang et al., 2025a) | Differential | ViT-S/16 | 81.38% | 81.11% | -0.27% | 4 |
| | | | ResNet-34 | 76.42% | 76.04% | -0.38% | 8 |
| | AdaFire (Wang et al., 2025) | Burst | ResNet-34 | 75.66% | 75.38% | -0.28% | 64 |
| | **TWE (ours)** | TWE | ResNet-34 | 75.53% | 75.51% | -0.02% | **3** |
| | | | VGG-16 | 74.33% | 74.32% | -0.01% | 4 |
| | | | ViT-S/16 | 81.51% | 81.59% | **+0.08%** | 5 |
| | | | ViT-L/16 | 83.81% | 83.72% | -0.09% | 5 |

[†] Ternary Spike is a direct training method that incorporates negative spikes.

## 4.3. TWE Neuron Model

We now summarize the proposed dynamics into a unified neuron model for TWE:

$$
\begin{cases}
o[t] = 2 \times u[t-1] + z[t], \\
s[t] = \mathbb{I}_{t \geq T_r} \cdot [H(o[t] - 2^{T_r-1}\theta) - H(-o[t] - 2^{T_r-1}\theta)], \\
u[t] = o[t] - 2^{T_r}\theta s[t].
\end{cases}
$$

**Hardware Feasibility.** The proposed model incurs minimal overhead on standard neuromorphic hardware. The recursive $\times 2$ operation corresponds to a single-bit left shift, which can be realized through simple wiring (connecting low bits of the membrane potential register to high bits of the adder input) without additional energy consumption. Signed spikes are natively supported on platforms such as

Intel Loihi 2, while custom implementations require only one additional sign bit per spike event. Notably, the spike magnitudes remain unmodulated, thereby preserving the event-driven nature of the network. These properties indicate that the proposed model requires only minimal modifications to standard IF neuron hardware, enabling efficient deployment on neuromorphic architectures.

## 5. Experiments

In this section, we conduct extensive experiments to evaluate the effectiveness of the proposed Temporal Weighted Encoding. We first compare TWE with existing encoding schemes on standard image classification benchmarks. Next, we analyze the energy consumption of TWE and investigate

the effects of temporal relaxation and threshold relaxation on its efficiency. We then perform ablation studies on these two mechanisms to examine their impact on task performance, which serves as an indicator of encoding accuracy. Furthermore, we empirically verify Theorem 4.4 under practical settings. Finally, we extend TWE to multiple tasks, including object detection, sentiment analysis, and classification on neuromorphic datasets, to evaluate its robustness under diverse input distributions.

**Experimental Setup.** We adopt the QCFS function (Bu et al., 2022; Hao et al., 2023) as the activation function for ANN training. QCFS function applies a learnable scaling factor to quantize activation values into $L$ discrete levels. During ANN-SNN conversion, the spike magnitude $\theta^l$ of each layer is set to the learned scaling factor, and the activations are encoded using $T = \lceil \log L \rceil$ timesteps. The choice of $T_r$ depends on the distribution of ANN activations. To achieve optimal performance, we set $T_r = 2$ on ImageNet and $T_r = 1$ for all other experiments. $T_r = 2$ is sufficient across all tested scenarios, including architectures with heavy-tailed activation distributions (e.g., ViTs on ImageNet). We encode network inputs by direct coding (Rueckauer et al., 2017; Hu et al., 2023; You et al., 2024). The real-valued activations of the first hidden layer are interpreted as constant currents, which are fed to the neurons of the subsequent layer.

## 5.1. Static Image Classification

We evaluate TWE on standard static image classification benchmarks, including CIFAR-10, CIFAR-100, and ImageNet. The results are summarized in Section 4.2.

On CIFAR-10 and CIFAR-100, TWE matches the accuracy of the original ANN using only three timesteps. In comparison, while TTFS-based methods (Stanojevic et al., 2024; 2023) can achieve high encoding precision, they typically require thousands of simulation steps to represent values accurately. Rate-base Fast-SNN (Hu et al., 2023) achieves low latency through aggressive quantization (e.g., 2-bit); however, it relies on post-conversion fine-tuning. Ternary Spike (Guo et al., 2024), a direct training method that also leverages negative spikes, achieves very low latency on CIFAR-10, but its accuracy remains below that of TWE.

On the large-scale ImageNet benchmark, the advantages of TWE become even more pronounced. TTFSFormer (Zhao et al., 2025) achieves near-lossless conversion for Transformer architectures but comes at the cost of large latency (thousands of steps). AdaFire (Wang et al., 2025) utilizes burst firing to mitigate conversion errors, yet it struggles to close the accuracy gap and offers no significant latency advantage. Differential coding (Huang et al., 2025a) successfully reduces timesteps, but suffers from larger con-

*Table 2.* Energy consumption estimation of VGG-16 on CIFAR-10, averaged over 10 inputs.

| Coding Scheme | ACs | MACs | Energy |
|---|---|---|---|
| $\mathbb{R}$ (ANN) | 0 | 313.60M | 1.4426mJ |
| TTFS (**Optimal**) | **120.53M** | **0**[†] | **0.1085mJ** |
| Rate | 295.10M | 12.85M | 0.3247mJ |
| **TWE** | 149.83M | 5.51M | 0.1602mJ |

[†] No MAC operations are involved in TTFS, as the network receives spiking inputs.

version loss and shows limited compatibility with standard CNNs. Although direct training approaches can operate at low latency on simpler datasets, scaling them to challenging tasks remains difficult. For example, Ternary Spike achieves 70.74% accuracy on ImageNet with 4 timesteps, whereas TWE attains 75.51% using 3 timesteps.

Overall, TWE consistently achieves a superior trade-off between accuracy preservation and encoding efficiency across diverse architectures.

## 5.2. Energy Consumption Analysis

In SNNs, each fired spike triggers an accumulate (AC) operation in all post-synaptic neurons. Therefore, the energy consumption is commonly approximated by the total number of spikes. Specifically, the energy consumption can be estimated as (Yang et al., 2024b; Suetake et al., 2023):

$$E_s = T \times E_{AC} \times \sum_l F^l \times \bar{r}^l$$

where $T$ denotes the number of timesteps, $E_{AC}$ represents the energy cost of an AC operation and $\bar{r}^l$ is the average firing rate of layer $l$. $F^l$ denotes the number of floating-point operations:

$$F^l = \begin{cases} (K^l)^2 \times W^l \times H^l \times C_{\text{in}}^l \times C_{\text{out}}^l & \text{for conv layer,} \\ C_{\text{in}}^l \times C_{\text{out}}^l & \text{for linear layer,} \end{cases}$$

where $K^l$ denotes the kernel size, $W^l$ and $H^l$ are the spatial dimensions of the output feature map, and $C_{\text{in}}^l$ and $C_{\text{out}}^l$ are the numbers of input and output channels, respectively. Since direct coding is adopted for the network input, the first layer performs multiply-accumulate (MAC) operations. This part of energy can be estimated as:

$$E_f = T \times E_{MAC} \times \sum_l F^l.$$

We benchmark the energy efficiency [2] of TWE against rate and TTFS coding on VGG-16, and the results are detailed

---

[2] Energy consumption estimation were performed based on https://github.com/iCGY96/syops-counter

*Table 3.* Performance comparison of TWE with rate coding across diverse tasks and datasets. For each task, we reproduce the SOTA rate-based ANN-SNN conversion methods (Hu et al., 2023; You et al., 2024). TWE consistently achieves comparable or improved SNN accuracy with reduced timesteps, demonstrating its robustness under diverse input modalities and distributions.

| Task | Dataset | Arch. | Coding Scheme | SNN Acc. | $\Delta$ Acc. $\uparrow$ | T $\downarrow$ |
|---|---|---|---|---|---|---|
| Neuromorphic Datasets | DVS128Gesture | ResNet-18 | Rate | 90.76% | -0.08% | 7 |
| | | | TWE | 90.89% | **+0.05%** | **3** |
| | CIFAR10-DVS | ResNet-18 | Rate | 78.06% | -0.29% | 7 |
| | | | TWE | 78.31% | **-0.04%** | **3** |
| Object Detection | VOC2007 | YOLOv2 (ResNet-34) | Rate | 73.43% | -1.84% | 7 |
| | | | TWE | 75.20% | **-0.07%** | **3** |
| Audio Classification | ESC-50 | ResNet-18 | Rate | 75.09% | -0.06% | 7 |
| | | | TWE | 75.14% | **-0.01%** | **3** |
| Sentiment Analysis | IMDB-MR | RoBERTa-B | Rate | 81.31% | -0.10% | 64 |
| | | | TWE | 81.36% | **-0.05%** | **5** |

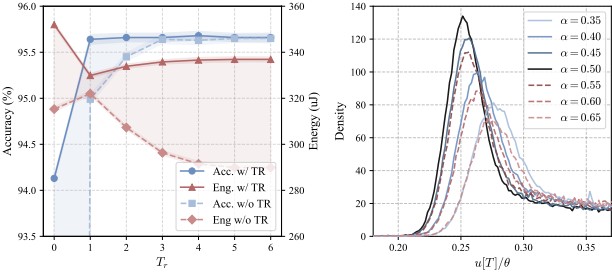

*Figure 4.* Left: Accuracy (left y-axis) and energy consumption (right y-axis) on CIFAR-10 using VGG-16 with $T = 6$. The relaxation parameter $T_r$ varies from 0 to 6. In the legend, "TR" stands for threshold relaxation. The proposed threshold relaxation method (solid lines) maintains high accuracy across the entire range of $T_r$, but the lowered threshold results in roughly 10% increase in energy consumption, representing a trade-off between accuracy and energy efficiency. Right: Distribution of the normalized residual membrane potential $u[T]/\theta$ under varying $\alpha \in [0.35, 0.65]$, obtained using ResNet-18 on CIFAR-10 when $T_r = 0$. The density plots illustrate the impact of the firing threshold on membrane potential statistics. The black curve represents the theoretically optimal value $\alpha = 0.5$, while other colored curves show the distributions for different $\alpha$ settings.

in Table 2. TTFS serves as the theoretical lower bound for energy consumption, as it enforces a single-spike constraint and eliminates MAC operations. Against this optimal baseline, TWE demonstrates competitive efficiency (0.1602 mJ), significantly outperforming standard rate coding. This efficiency stems from the reduced timesteps.

As both temporal relaxation and threshold relaxation modify the output spiking dynamics, we further investigate their impact on energy consumption. To allow sufficient variation of $T_r$, $T$ is set to 6. The results are shown in Figure 4 (Left).

Since threshold relaxation explicitly encourages spike emission, it leads to a slight increase in energy consumption (around 10%). Another observation is that as $T_r$ approaches $T$, the energy consumption converges to a stable value. This is beacuse the output spikes $s[t + T_r]$ converges to the ideal representation $s^*[t]$ in this condition.

**5.3. Ablation and Theoretical Validation**

To validate the effectiveness of the proposed threshold relaxation strategy, we conducted an ablation study on the temporal relaxation parameter $T_r$ using VGG-16 on CIFAR-10 with a total timestep $T = 6$. The results are illustrated in Figure 4 (Right). As indicated by the solid blue curve, the accuracy remains high across the entire range of $T_r$. Notably, in the challenging online coding scenario where $T_r = 0$, threshold relaxation drastically boosts the accuracy from 13.16% to a competitive 94.13%. This large performance margin highlights the critical role of this mechanism in enabling effective information transmission at the early stage.

Furthermore, we investigate the rationality of the threshold scaling factor $\alpha$. Specifically, we analyze the distribution of the residual membrane potential $u[T]$ using a ResNet-18 model on CIFAR-10. As visualized in Figure 4, the setting $\alpha = 0.5$ yields the most stable distribution with a concentrated peak. In contrast, deviating from this value (either $\alpha < 0.5$ or $\alpha > 0.5$) leads to skewed distributions, which implies increased enccoding error.

An interesting observation is that when two values $\alpha_1$ and $\alpha_2$ are symmetric with respect to $0.5$, the resulting distributions of $u[T]$ are highly similar. We visualize such pairs using solid and dashed curves of the same color in Figure 4. This symmetry is consistent with the formula-

tion in Theorem 4.2, where the dynamics are governed by $\beta = \max(\alpha, 1 - \alpha)$, suggesting that this quantity indeed plays a central role in shaping the membrane potential dynamics.

### 5.4. Generalization to Diverse Tasks

To evaluate the robustness and generality of TWE under diverse input modalities and data distributions, we further extend our experiments beyond static image classification. A wide range of tasks are considered, including object detection, sentiment analysis, neuromorphic event-based recognition, and audio classification. For each task, we benchmark our method against SOTA rate coding implementations (Hu et al., 2023; You et al., 2024) using identical ANN weights.

The results are summarized in Section 5.2. Across all evaluated tasks and datasets, TWE consistently achieves comparable or superior SNN accuracy while requiring fewer timesteps. For instance, on the object detection task with YOLOv2 on VOC2007, TWE improves the detection accuracy from 73.43% to 75.20% while reducing the required timesteps from 7 to 3. Similarly, on neuromorphic datasets such as DVS128Gesture, TWE not only reduces the encoding timesteps by more than half, but also slightly improves the accuracy. These results indicate that the temporal weighted encoding is robust to different input distributions and model architectures.

## 6. Conclusion

In this paper, we addressed the fundamental bottleneck of encoding efficiency in ANN-SNN conversion by proposing Temporal Weighted Encoding (TWE). Drawing inspiration from digital binary representation, TWE unlocks the exponential capacity of the spike encoding space through a simple recursive integration mechanism.

Under this weighted encoding scheme, we identified the inherent temporal mismatch challenge and systematically resolved it by temporal relaxation and threshold relaxation mechanisms. Supported by rigorous theoretical analysis, these strategies effectively bridges the lagging inputs and ideal outputs, enabling fast and accurate encoding. Extensive experiments across diverse architectures and benchmarks demonstrate that TWE achieves negligible conversion loss with significantly fewer timesteps compared to existing encoding schemes, offering a scalable and hardware-friendly solution for next-generation neuromorphic computing.

## Acknowledgements

This work was supported by Zhejiang Province's Leading Talent Project in Science and Technology Innovation (2023R5204), National Natural Science Foundation of China (Grant No. 62274142), and Sino-German Mobility Programme (Grant No. M-0499).

## Impact Statement

This paper presents work whose goal is to advance the field of Machine Learning. There are many potential societal consequences of our work, none which we feel must be specifically highlighted here.

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

# A. Proofs

## A.1. Proof of Theorem 4.2

*Proof of Theorem 4.2, Part 1.* We first establish the matrix form of the encoding condition under no delay, and then show how introducing $T_r$ transforms this system.

**Base case ($T_r = 0$).** Under the recursive integration dynamics $o[t] = 2u[t-1] + z[t]$, the total weighted input accumulated over $T$ timesteps is

$$I(z) = \sum_{t=1}^{T} 2^{T-t} z[t].$$

For the neuron to produce the target output $s^*[t]$, this total input must satisfy

$$R_{\text{TWE}}(s^*) \leq I(z) < R_{\text{TWE}}(s^*) + v_{th},$$

where the lower bound ensures sufficient accumulation for the target pattern and the upper bound prevents overshooting. Considering the threshold $v_{th}$ is negligible compared to $I(z)$, this simplifies to $I(z) \approx R_{\text{TWE}}(s^*)$.

To see the local firing constraints, temporarily ignore the reset after each spike emission. At step $t$, the accumulated membrane potential is the weighted sum $\sum_{\tau=1}^{t} 2^{t-\tau} z[\tau]$. If $s^*[t] = 1$, this potential must exceed the threshold $v_{th} = \theta$; equivalently,

$$\sum_{\tau=1}^{t} 2^{t-\tau} z[\tau] \geq \sum_{\tau=1}^{t} 2^{t-\tau} s^*[\tau] \cdot \theta.$$

The right-hand side is precisely the partial binary representation of the target pattern $s^*$ up to step $t$, scaled by $\theta$. When $s^*[t] = 0$, no such constraint is needed, because the total input being tied to $R_{\text{TWE}}(s^*)$ implicitly prevents premature firing.

Defining the prefix sum $S_t = \sum_{\tau=1}^{t} 2^{t-\tau} z[\tau]$, the encoding condition is equivalent to the following matrix inequality:

$$\begin{bmatrix} 1 & 0 & 0 & \cdots & 0 \\ 2 & 1 & 0 & \cdots & 0 \\ 4 & 2 & 1 & \cdots & 0 \\ \vdots & \vdots & \vdots & \ddots & \vdots \\ 2^{T-1} & 2^{T-2} & 2^{T-3} & \cdots & 1 \end{bmatrix} \begin{bmatrix} z[1] \\ z[2] \\ z[3] \\ \vdots \\ z[T] \end{bmatrix} \geq \begin{bmatrix} b_1 \\ b_2 \\ b_3 \\ \vdots \\ b_T \end{bmatrix},$$

or compactly $S_t \geq b_t$ for $t = 1, \ldots, T$, where $b_t$ corresponds to the partial binary value of $s^*$ up to step $t$ (appropriately scaled by $\theta$), together with the box constraints $z[\tau] \in [-Z, Z]$.

**Step 1: Algebraic simplification.** The prefix sums satisfy the recursive relation $S_t = 2S_{t-1} + z[t]$ (with $S_0 = 0$). Under this change of variables, the original matrix inequality collapses into the simple half-space constraints $S_t \geq b_t$.

**Step 2: Introducing delay $T_r$.** To introduce temporal relaxation with delay $T_r > 0$, we apply the following linear change of variables to the base system:

$$z'[1] = \frac{1}{2^{T_r}} S_{T_r} = \frac{1}{2^{T_r}} \sum_{\tau=1}^{T_r} 2^{T_r - \tau} z[\tau], \qquad z'[t] = \frac{1}{2^{T_r}} z[t + T_r - 1] \quad \text{for } t = 2, \ldots, T - T_r + 1.$$

This transformation folds the first $T_r$ input variables into a single combined variable $z'[1]$, while scaling the remaining variables by $2^{-T_r}$. Geometrically, this corresponds to integrating out the first $T_r$ dimensions of the original feasible space. Applying the same inequality structure to $z'$, the $k$-th constraint becomes

$$\frac{1}{2^{T_r}} S_{T_r + k} \geq b_k, \quad k = 1, \ldots, T - T_r.$$

Under the scaling condition $b_m \approx 2b_{m-1}$, which holds for typical target patterns $s^*$, we have $b_{T_r + k} \approx 2^{T_r} b_k$. Thus the delayed constraints are equivalent to

$$S_{T_r + k} \geq b_{T_r + k}, \quad k = 1, \ldots, T - T_r.$$

Geometrically, the feasible space with delay $T_r$ is identical to the original feasible space but with the first $T_r$ half-space constraints $(S_1 \geq b_1, \ldots, S_{T_r} \geq b_{T_r})$ removed.

**Step 3: Slack variables and boundary mechanics.** Introduce slack variables $y_t = S_t - b_t$, so that the constraints become $y_t \geq 0$. Using $z[t] = S_t - 2S_{t-1}$ and the approximation $b_t \approx 2b_{t-1}$, the box constraints $|z[t]| \leq Z$ map into

$$|2y_{t-1} - y_t| \leq Z \implies y_{t-1} \in \left[ \frac{y_t}{2} - \frac{Z}{2}, \frac{y_t}{2} + \frac{Z}{2} \right].$$

Since the Jacobian determinant of the linear transformation from $z$ to $y$ is $\pm 1$, volume ratios are preserved.

**Step 4: Dimensional volume scaling.** In the $y$-space, each variable $y_{t-1}$ is confined to an interval of width $Z$. For the original feasible space $\mathcal{Z}(0, s^*)$, the non-negativity constraints $y_t \geq 0$ truncate the effective interval to approximately $[0, Z/2]$, halving the available phase space along each constrained dimension. In contrast, the delayed feasible space $\mathcal{Z}(T_r, s^*)$ drops the first $T_r$ constraints, so $y_1, \ldots, y_{T_r}$ are free to span their full natural widths. Consequently,

$$\frac{\mathrm{Vol}(\mathcal{Z}(T_r, s^*))}{\mathrm{Vol}(\mathcal{Z}(0, s^*))} \approx 2^{T_r}. \qquad \square$$

*Proof of Theorem 4.2, Part 2.* When $T_r = T$, the output is delayed until all input currents have been fully accumulated. In this case, the neuron can compute the total weighted input $I(z) = \sum_{t=1}^{T} 2^{T-t} z[t]$ before making any firing decision. Since TWE can represent any integer value in $[0, 2^T - 1]$, there always exists a spike pattern $s^*[t]$ such that $R_{\mathrm{TWE}}(s^*) = I(z)$. Therefore, the feasible space covers all valid inputs and ideal encoding is achieved. $\qquad \square$

### A.2. Proof of Proposition 4.3

*Proof of Proposition 4.3.* From the integration dynamics in (1) and the firing dynamics in (3), the membrane potential before reset obeys $o[t] = 2u[t-1] + z[t]$. Under the assumption $v_{th} = \theta$, when a spike is emitted the potential is reset by subtracting $2^{T_r} \theta\, s[t]$, so the post-reset potential satisfies

$$u[t] = o[t] - 2^{T_r} \theta\, s[t].$$

Together with the initial condition $u[0] = 0$ and the fact that $s[t] = 0$ for $t < T_r$ (enforced by the indicator in (3)), unrolling the recursion over $T + T_r$ timesteps yields

$$u[T + T_r] = \sum_{t=1}^{T+T_r} 2^{T+T_r-t} z[t] - 2^{T_r} \theta \sum_{t=1}^{T+T_r} 2^{T+T_r-t} s[t].$$

Since the input $z$ has only $T$ components, $z[t] = 0$ for $t > T$. Hence the first term simplifies to

$$\sum_{t=1}^{T+T_r} 2^{T+T_r-t} z[t] = \sum_{t=1}^{T} 2^{T+T_r-t} z[t] = 2^{T_r} \sum_{t=1}^{T} 2^{T-t} z[t] = 2^{T_r} I(z).$$

For the second term, the constraint $s[t] = 0$ for $t < T_r$ gives

$$\theta \sum_{t=1}^{T+T_r} 2^{T+T_r-t} s[t] = \theta \sum_{t=T_r+1}^{T+T_r} 2^{T+T_r-t} s[t] = \theta \sum_{t'=1}^{T} 2^{T-t'} s[t' + T_r] = R_{\mathrm{TWE}}(s).$$

Substituting these back, we obtain

$$u[T + T_r] = 2^{T_r} I(z) - 2^{T_r} R_{\mathrm{TWE}}(s).$$

By definition of the ideal encoding, $R_{\mathrm{TWE}}(s^*) = I(z)$. Therefore,

$$u[T + T_r] = 2^{T_r} \big( R_{\mathrm{TWE}}(s^*) - R_{\mathrm{TWE}}(s) \big),$$

which rearranges to the claimed result:

$$R_{\mathrm{TWE}}(s) - R_{\mathrm{TWE}}(s^*) = -\frac{1}{2^{T_r}} u[T + T_r]. \qquad \square$$

## A.3. Proof of Theorem 4.4

*Proof of Theorem 4.4.* The proof consists of three parts. We first establish the boundedness of the residual membrane potential for general $\alpha$, then prove the convergence to a uniform invariant measure, and finally derive the optimal variance.

**Part I: Boundedness of the invariant set.** Consider the neuron dynamics with $T_r = 0$:

$$o[t] = 2u[t-1] + z[t], \quad s[t] = H(o[t] - \alpha\theta) - H(-o[t] - \alpha\theta), \quad u[t] = o[t] - \theta s[t].$$

We show that $\Omega = [-\beta\theta, \beta\theta]$ is a strictly invariant set, where $\beta = \max(\alpha, 1 - \alpha)$.

Assume $u[t-1] \in [-\beta\theta, \beta\theta]$ and $|z[t]| \leq Z$. We distinguish three cases according to the value of $s[t]$.

**Case 1** ($s[t] = 0$). No spike is emitted, so $-\alpha\theta < o[t] < \alpha\theta$. Since $u[t] = o[t]$ and $\alpha \leq \beta$, we immediately have $u[t] \in (-\alpha\theta, \alpha\theta) \subset [-\beta\theta, \beta\theta]$.

**Case 2** ($s[t] = 1$). A positive spike is triggered, which requires $o[t] \geq \alpha\theta$. The post-reset potential is $u[t] = o[t] - \theta$. For the upper bound,

$$u[t] \leq 2\beta\theta + Z - \theta = (2\beta - 1)\theta + Z.$$

By the theorem assumption $\theta \geq \frac{Z}{1-\beta}$, we have $Z \leq (1 - \beta)\theta$, hence

$$u[t] \leq (2\beta - 1)\theta + (1 - \beta)\theta = \beta\theta.$$

For the lower bound,

$$u[t] \geq \alpha\theta - \theta = -(1 - \alpha)\theta \geq -\beta\theta,$$

where the last inequality follows from $\beta = \max(\alpha, 1 - \alpha) \geq 1 - \alpha$.

**Case 3** ($s[t] = -1$). A negative spike is triggered, which requires $o[t] \leq -\alpha\theta$. The post-reset potential is $u[t] = o[t] + \theta$. For the lower bound,

$$u[t] \geq -2\beta\theta - Z + \theta = (1 - 2\beta)\theta - Z \geq (1 - 2\beta)\theta - (1 - \beta)\theta = -\beta\theta.$$

For the upper bound,

$$u[t] \leq -\alpha\theta + \theta = (1 - \alpha)\theta \leq \beta\theta.$$

Combining the three cases, if $u[t-1] \in [-\beta\theta, \beta\theta]$ then $u[t] \in [-\beta\theta, \beta\theta]$. By induction, $\Omega = [-\beta\theta, \beta\theta]$ is a global attractor and strictly invariant set.

**Part II: Convergence to a uniform invariant measure.** To analyze the limiting distribution, we first consider the deterministic skeleton obtained by setting $z[t] = 0$. The core map $T : \Omega \to \Omega$ is

$$T(u) = \begin{cases} 2u, & |u| < \frac{\alpha\theta}{2}, \\ 2u - \theta, & u \geq \frac{\alpha\theta}{2}, \\ 2u + \theta, & u \leq -\frac{\alpha\theta}{2}. \end{cases}$$

The evolution of the probability density is governed by the Frobenius-Perron operator $\mathcal{P}$:

$$(\mathcal{P}f)(y) = \sum_{x \in T^{-1}(y)} \frac{f(x)}{|T'(x)|} = \frac{1}{2} \sum_{x \in T^{-1}(y)} f(x),$$

since $|T'(x)| = 2$ almost everywhere.

When $\alpha = 0.5$, the three branches have ranges $(-\theta/2, \theta/2)$, $[-\theta/2, 0]$, and $[0, \theta/2]$, respectively. These intervals tessellate $\Omega$ without triple overlap: every $y \in \Omega$ has exactly two pre-images (except at the boundaries). Substituting the constant density $f^*(u) = 1/\theta$ gives

$$(\mathcal{P}f^*)(y) = \frac{1}{2} \cdot 2 \cdot \frac{1}{\theta} = \frac{1}{\theta} = f^*(y),$$

so $f^*$ is a fixed point. Because $T$ is piecewise expanding with slope $2 > 1$, the Lasota-Yorke theorem guarantees that this invariant measure is unique and ergodic. Consequently, for $\alpha = 0.5$,

$$\lim_{t \to \infty} f_t(u) = \mathcal{U}[-\beta\theta, \beta\theta] = \mathcal{U}\left[-\frac{\theta}{2}, \frac{\theta}{2}\right].$$

For general $\alpha \in (0, 1)$, the bounded random input $z[t]$ acts as a dither signal: at each step it perturbs the deterministic trajectory, smoothing the density and accelerating convergence. Under this mixing condition the limiting distribution is well approximated by the uniform measure $\mathcal{U}[-\beta\theta, \beta\theta]$ supported on the invariant set.

**Part III: Variance minimization.** Since the limiting distribution is uniform on $[-\beta\theta, \beta\theta]$, its variance is

$$\text{Var}(u) = \mathbb{E}[u^2] = \int_{-\beta\theta}^{\beta\theta} u^2 \cdot \frac{1}{2\beta\theta} \, du = \frac{(\beta\theta)^2}{3}.$$

Since $\beta = \max(\alpha, 1 - \alpha) \geq 0.5$ with equality if and only if $\alpha = 0.5$, the variance is minimized at $\alpha = 0.5$, yielding

$$\min_\alpha \text{Var}(u) = \frac{\theta^2}{12}. \qquad \qquad \square$$

