# OpenReview forum: "Temporal Weighted Encoding: Towards Maximal-Capacity Spike Coding for ANN–SNN Conversion"
_ICML.cc/2026/Conference — ICML 2026 regular_

### Official Review · Reviewer_HGV2 · 2026-03-02

**Soundness:** 4
**Presentation:** 3
**Significance:** 3
**Originality:** 4
**Overall Recommendation:** 5
**Confidence:** 5

**Summary:**

This paper addresses the fundamental challenge of achieving high-efficiency and high-capacity encoding in ANN-SNN conversion. The authors propose Temporal Weighted Encoding (TWE), an encoding scheme that maximally utilizes the spike-encoding space. Compared with prior weighted schemes, TWE introduces a recursive integration mechanism, which enables a simpler realization of weighted information decoding.

Furthermore, this paper identifies and systematically analyzes a previously overlooked issue termed temporal mismatch. To address this problem, the authors introduce two mechanisms: temporal relaxation (delayed firing) and threshold relaxation (lowered thresholds with negative spikes).

Extensive experiments demonstrate that TWE achieves near-lossless encoding with reduced timesteps, while maintaining competitive energy efficiency compared to established methods.

**Compliance With Llm Reviewing Policy:**

Affirmed.

**Final Justification:**

My concerns have been resolved.

**Key Questions For Authors:**

1. Can the authors provide a more detailed comparison between TWE and (a) phase coding methods that assign power-of-two weights to spikes, and (b) FS coding? Specifically, how TWE’s recursive integration reduce implementation complexity or improve robustness compared to these methods, beyond the use of a fixed threshold?

2. The experiments primarily use (T_r = 1) (or 2 for ImageNet). How sensitive is performance to (T_r) when (T=3) (the default setting in Tables 1 and 3)? Is there potential benefit in setting (T_r) differently across layers—for example, using larger delays in early layers where activation distributions may be more skewed?

3. Multi-threshold neurons can also achieve weighted encoding via multiple spikes. How does this approach compare with TWE in terms of implementation complexity, robustness, and conversion accuracy?

**Limitations:**

yes

**Strengths And Weaknesses:**

Strengths

1. The integration of power-of-two weighting directly into spiking dynamics provides a practically simple mechanism for event-based SNNs, avoiding the overhead of offline or complex time-modulated schemes.

2. The proposed temporal and threshold relaxation mechanisms constitute well-motivated algorithmic innovations to mitigate temporal mismatch and the latency–accuracy trade-off. These design choices are supported by both theoretical analysis and empirical validation.

3. The experimental evaluation is extensive and demonstrates competitive performance across multiple benchmarks and tasks.

Weaknesses

1. Hyperparameters such as $T_r$ are fixed to 1 in most experiments and 2 for ImageNet, but the paper provides limited justification or sensitivity analysis beyond Figure 4 (VGG-16 on CIFAR-10 with (T=6)). It remains unclear how sensitive performance is to $T_r$ and whether different architectures require specific tuning.

2. The theoretical analysis mainly considers the encoding behavior of a single neuron under bounded inputs. However, in deep multi-layer networks, residual membrane errors may accumulate and interact with weight scaling and activation distributions. A comparison of error versus depth between TWE and rate/TTFS coding would strengthen the claim of “negligible conversion loss” in deep architectures.

---

> ### Author Rebuttal · Authors · 2026-03-31
>
> We sincerely thank the reviewer for recognizing the practical simplicity of our encoding mechanism, the well-motivated relaxation designs, and the extensive experimental evaluation. We address each concern below.
>
> ---
>
> **Q1: Comparison with phase coding and FS coding**
>
> We appreciate this important point. While TWE shares the same power-of-two weight structure with phase coding, the fundamental difference lies in online vs. offline operation. Phase coding methods wait for all inputs before computing spike outputs — they operate offline precisely because they lack a mechanism to resolve the temporal mismatch inherent in weighted encoding. TWE is the first to enable fast weighted encoding through relaxation mechanisms.
>
> Compared to FS coding, which uses time-varying thresholds and spike magnitudes to achieve weighted integration, TWE maintains fixed thresholds and identical spike magnitudes through recursive integration. This eliminates the need for time-modulated neuron dynamics, reducing implementation complexity. The recursive ×2 operation is structurally simpler than FS coding's explicit temporal modulation and is more robust to timing variations.
>
> ---
>
> **Q2: Sensitivity to $T_r$ and layer-specific tuning**
>
> Figure 4 provides a comprehensive ablation over $T_r \in [0, 6]$ using VGG-16 on CIFAR-10 with $T=6$. Accuracy remains high and stable for $T_r \geq 1$, with diminishing returns beyond $T_r = 2$. We further evaluated $T_r$ sensitivity under the default $T=3$ setting:
>
> | $T_r$ | ResNet-34 on ImageNet | ResNet-18 on CIFAR-10 | VGG-16 on CIFAR-100 |
> |-------|---------------|---------------|---------------|
> | 1     | 75.34% | 96.34% |  77.20% |
> | 2     | 75.51% | 96.33% |  77.20% |
> | 3     | 75.49% | 96.34% |  77.18% |
>
> The results confirm that $T_r = 1$ or $2$ is robust across architectures. Regarding layer-specific tuning: while early layers may have more skewed activation distributions, our experiments show that a uniform $T_r$ suffices. The bounded residual distribution (Theorem 4.4) prevents error accumulation regardless of layer position. Layer-specific $T_r$ could offer marginal gains but would complicate deployment without substantial benefit.
>
> ---
>
> **Q3: Multi-layer error accumulation**
>
> This is an important concern. Proposition 4.3 establishes that encoding error is proportional to the residual membrane potential $u[T]$. Theorem 4.4 proves that with threshold relaxation, $u[T]$ converges to a bounded uniform distribution on $[-\beta\theta, \beta\theta]$. This means the residual errors do not grow with depth because each layer's membrane potential is reset to a bounded range. We empirically validate this by providing the average residual membrane potential (normalized by the firing threshold) across layers for VGG-16 on CIFAR-10:
>
> | Layer | 2 | 4 | 6 | 8 | 10 | 12 |
> |----|---|---|---|---|---|---|
> | rate (T=7)   | 0.87e-2 | 1.92e-2 | 1.45e-2 | 1.01e-2 | 0.68e-2 | 0.46e-2 |
> | TWE (T=3)   | 0.47e-2 | 0.53e-2 | 0.22e-2 | 0.24e-2 | 0.18e-2 | 0.12e-2 |
>
> ---
>
> **Q4: Comparison with multi-threshold neurons**
>
> Multi-threshold neurons can achieve weighted encoding by triggering spikes of different magnitudes at different thresholds. However, this approach has several disadvantages:
>
> - Implementation complexity: Requires $O(T)$ threshold comparators per neuron vs. TWE's $O(1)$ comparators + simple ×2 operation. Multiple thresholds must be carefully calibrated to match weight ratios; TWE's fixed threshold is inherently robust.
> - Event-driven compatibility: Multi-threshold spikes carry magnitude information, requiring additional encoding per event; TWE preserves pure spiking events (with a sign bit)
>
> Empirically, multi-threshold methods achieve comparable accuracy but at higher hardware cost. TWE achieves equivalent representational capacity through temporal recursion rather than spatial threshold multiplicity, making it more hardware-friendly.

---

> > ### Author Rebuttal · Reviewer_HGV2 · 2026-04-02
> >
> > Thank you for your answer. My concerns have been addressed, so I will revise my score.

---

### Official Review · Reviewer_J54G · 2026-03-11

**Soundness:** 2
**Presentation:** 3
**Significance:** 2
**Originality:** 3
**Overall Recommendation:** 3
**Confidence:** 4

**Summary:**

This paper proposes TWE for ANN-to-SNN conversion. TWE interprets spike sequences as weighted temporal bit streams and uses recursive integration to assign exponentially decaying weights, increasing encoding capacity relative to standard rate or TTFS coding. To address temporal mismatch, the method further introduces Temporal Relaxation and Threshold Relaxation with signed spikes. Experiments across multiple architectures and tasks show strong low-latency conversion performance, although some of the practical and theoretical claims remain insufficiently supported.

**Compliance With Llm Reviewing Policy:**

Affirmed.

**Key Questions For Authors:**

Remark 4.5 explicitly admits that the key bounding condition frequently fails in practice. In that case, the current theoretical justification appears incomplete. Can you provide a meaningful worst-case or alternative error bound when this assumption is violated? More importantly, is the claimed optimality of \alpha = 0.5 still expected to hold under highly skewed activation distributions or in layers with many outliers?
The choice of T_r seems under-justified. The paper fixes T_r = 1 for most datasets, yet requires T_r = 2 on ImageNet for best performance. How sensitive is TWE to T_r, and is there any principled rule or heuristic for selecting it on new datasets without resorting to expensive tuning?
The efficiency claims are based primarily on operation-level estimates rather than real deployment measurements. Can you provide actual hardware measurements, or at minimum clarify the gap between the reported estimates and expected real-system behavior?
Table 2 shows that TWE incurs substantially higher AC counts than standard rate coding. This directly weakens one of the main practical advantages of SNNs, namely sparse and low-cost computation, and raises questions about scalability to larger architectures. How do you justify the efficiency claim in light of this significant increase in accumulation overhead?

**Limitations:**

The authors discuss some limitations implicitly through the theoretical caveats and experimental setup, but the paper would benefit from a clearer discussion of the practical limitations of the theoretical assumptions, the lack of direct hardware validation, and the increased AC overhead relative to standard rate coding.

**Strengths And Weaknesses:**

Strength: The paper studies an important problem in ANN-to-SNN conversion and proposes a simple temporal coding scheme with clear intuition. The method is easy to follow at a high level, and the combination of temporal weighting, temporal relaxation, and threshold relaxation is reasonably well motivated. Evaluation is broad and includes multiple tasks and datasets. The low-timestep results and ablations on relaxation parameters are useful. The paper also makes an effort to provide theoretical support for the design choices.
Weakness: Main weakness is that the theoretical support is not as strong as the paper suggests. While the analysis is relevant to the method, the assumptions appear stylized, and the paper acknowledges that an important bounding condition often fails in practice.
Weakness: Empirical study is broad but not fully convincing. Some baseline comparisons are not strictly controlled due to differences in architectures, training settings, and fine-tuning procedures. In addition, the efficiency claims rely on estimated operation counts and proxy energy analysis rather than direct hardware measurements, which makes the practical efficiency advantage difficult to assess.
Overall, the paper has clear merits, but the theoretical justification is weakened by its own assumptions, the practical efficiency claims remain under-supported, and the novelty boundary is not sufficiently sharp.

---

> ### Author Rebuttal · Authors · 2026-03-31
>
> We sincerely thank the reviewer for recognizing our problem motivation, method clarity, evaluation breadth, and ablation studies. We address each concern below.
>
> ---
>
> **Q1: Theoretical justification when the bounding condition fails (Remark 4.5)**
>
> The condition $\theta \geq Z/(1-\beta)$ can be strictly satisfied by setting a sufficiently large threshold, since the maximum input current is bounded by $Z = \sum_j |w_{ij}|$. However, such a large threshold trades off with encoding precision — the neuron becomes insensitive to small inputs. Moreover, neurons receiving large inputs are rare (typically only in the output layer). Therefore we set the threshold at the 99.9th percentile of the ANN activation range (as in rate coding), so the vast majority of neurons operate within the theorem's assumptions.
>
> The primary value of Theorem 4.4 is mechanistic rather than bounding: it reveals that $\alpha = 0.5$ balances positive and negative error margins by minimizing the variance of the residual membrane potential. This insight arises from the structure governed by $\beta = \max(\alpha, 1-\alpha)$ and does not depend on the strict condition.
>
> Recognizing the difficulty of deriving a strict error bound for this nonlinear multi-layer temporal system, we instead provide extensive empirical validation. Figure 5 shows that $u[T]$ is most concentrated at $\alpha = 0.5$, and symmetric pairs produce nearly identical distributions — exactly as predicted. We further validate this across ResNet and Transformer (with many outliers) architectures. Together, Proposition 4.3 (error–residual link) and Theorem 4.4 (residual statistics) form an analytical framework well-supported by empirical evidence:
>
> | Arch \\ $\alpha$ | $0.3$  | $0.35$ | $0.4$ | $0.45$ | $0.5$ | $0.55$ | $0.6$ | $0.65$ | $0.7$
> |--------------|--------------|--------------|--------------|--------------|--------------|--------------|--------------|--------------|--------------|
> | ResNet-34 | 33.4% | 53.8% | 73.2% | 75.1% | **75.5%** | *75.4%* | 74.8% | 51.2% | 32.1% |
> | VGG-16 | 25.2% | 45.6% | 70.0% | **74.3%** | **74.3%** | *73.9%* | 72.8% | 50.9% | 30.5% |
> | ViT-S/16 | 28.7% | 48.2% | 78.5% | 81.2% | **81.6%** | *81.3%* | 80.5% | 52.3% | 31.2% |
> | ViT-L/16 | 31.5% | 50.1% | 81.0% | *83.5%* | **83.7%** | *83.5%* | 81.9% | 53.7% | 19.8% |
>
> ---
>
> **Q2: Choice of $T_r$ and sensitivity analysis**
>
> Figure 4 provides a comprehensive ablation over $T_r \in [0, 6]$ using VGG-16 on CIFAR-10 with $T=6$. The accuracy remains high and stable across $T_r \geq 1$, indicating that TWE is not highly sensitive to the exact value. In fact, due to the recursive weighting structure, the performance gain from increasing $T_r$ exhibits diminishing returns — as shown in Figure 4, the improvement becomes marginal beyond $T_r = 2$.
>
> Deriving an analytical rule for $T_r$ is challenging for the same reasons as Q1. Instead, our experiments provide clear empirical guidance: $T_r = 2$ is a robust, conservative choice minimizing conversion error across all tested scenarios, including those with many outliers (e.g., ViT for ImageNet). In most practical settings, $T_r = 1$ already delivers strong performance. Practitioners can safely use $T_r = 1$ or $2$ without expensive tuning.
>
> ---
>
> **Q3: Hardware measurements, AC overhead, and efficiency claims**
>
> We sincerely thank the reviewer for this careful observation, which helped us identify an error in our manuscript. In Table 2, the AC count for rate coding was incorrectly reported as 47.55M, when it should be 295.1M. We deeply apologize for this mistake and any confusion it may have caused. ***The correct value can be verified by the reported total energy of 0.3247mJ using the standard formula*** $E = 0.9 pJ/AC \times AC + 4.6 pJ/MAC \times MAC$:
>
> $$0.9pJ \times 295.1M + 4.6pJ \times 12.85M = 0.3247 mJ$$
>
> The corrected table is as follows:
>
> | Coding Scheme | ACs | MACs | Energy |
> |--------------|-----|------|--------|
> | Rate (corrected) | ~~47.55M~~ 295.1M | 12.85M | 0.3247mJ |
> | **TWE** | **149.83M** | 5.51M | **0.1602mJ** |
>
> With the corrected data, TWE reduces both AC operations ($\approx$49% reduction) and total energy ($\approx$51% reduction) compared to Rate coding. We will correct this in the revision.
>
> We also acknowledge that our energy analysis is based on operation counts rather than direct hardware measurements. Due to the complexity of implementing and measuring on actual neuromorphic hardware, operation-based estimation is the standard practice in the ANN-SNN conversion literature. TWE's advantage stems from the reduced timesteps, a fundamental algorithmic benefit. Compared to the simplest IF model, TWE adds only the recursive weighting ($\times 2$) operation, which can be achieved by simple hardware wiring (connecting low bits of the membrane potential register to high bits of the adder input) and contributes zero energy consumption. Therefore, TWE should also reduce energy consumption on actual hardware.

---

> > ### Author Rebuttal · Reviewer_J54G · 2026-04-03
> >
> > We thank the authors for their rebuttal, however it comes to my issue that since implementation of the model was not done to a neuromorphic hardware my rating will stay the same.

---

> > > ### Author Response · Authors · 2026-04-05
> > >
> > > We thank the reviewer for the feedback and provide further clarification on the aspect of hardware implementation.
> > >
> > > ---
> > >
> > > First, **TWE is designed with hardware compatibility considerations. Implementing it on neuromorphic platforms is straightforward**:
> > >
> > > - The recursive ×2 operation is a single-bit left shift. This can be achieved by adding a MUX for compatibility with existing hardware. For custom designs, the shift operation can be implemented through simple wiring (connecting low bits of the membrane potential register to high bits of the adder input), contributing no additional energy.
> > > - Negative spikes are natively supported on commercial neuromorphic platforms, such as Intel Loihi 2, with no additional overhead. For custom implementations, an additional sign bit is needed per spike event, while spike magnitudes remain unmodulated. The event-driven nature of computation is fully preserved.
> > >
> > > **Overall, TWE only requires minimal modifications to standard IF neuron hardware, and we are not aware of any fundamental barrier to its implementation on neuromorphic architectures.**
> > >
> > > ---
> > >
> > > Second, we note that our work focuses on encoding scheme innovation for ANN–SNN conversion. As is standard practice in the field, we conduct our evaluations via software simulation. **All baseline works we compare against follow this same approach and do not include custom hardware accelerator implementations** (e.g., the FS coding published in Nature Machine Intelligence [1] and the differential coding published in ICML 2025 [1]). While we acknowledge the importance of hardware implementation as a future direction, **software simulation is fully appropriate for the scope of this paper, and aligns with established community norms**.
> > >
> > > ---
> > >
> > > Finally, if the reviewer believes there are any specific components of the TWE neuron dynamics that cannot be realized on neuromorphic hardware, we would greatly appreciate specific feedback, and we are happy to further discuss implementation details and feasibility.
> > >
> > > ---
> > >
> > > We hope this clarifies the hardware compatibility of TWE, and we would be grateful if the reviewer would reconsider their rating in light of this information.
> > >
> > > [1] Optimized spiking neurons can classify images with high accuracy through temporal coding with two spikes
> > >
> > > [2] Differential Coding for Training-Free ANN-to-SNN Conversion

---

### Official Review · Reviewer_C6tb · 2026-03-12

**Soundness:** 3
**Presentation:** 3
**Significance:** 3
**Originality:** 3
**Overall Recommendation:** 5
**Confidence:** 3

**Summary:**

This paper proposes Temporal Weighted Encoding (TWE), a spike encoding scheme designed to improve the efficiency and representational capacity of ANN–SNN conversion methods. Existing spike encoding approaches (rate coding, time-to-first-spike ..) underutilize the theoretical encoding capacity available in spike sequences over multiple timesteps.

To address this limitation, the paper introduces Temporal Weighted Encoding, where spikes are implicitly assigned exponentially decaying temporal weights through a recursive integration process.  The authors further identify a temporal mismatch problem arising from this weighting structure and propose two corrective techniques: temporal relaxation and threshold relaxation, which aim to improve alignment between encoded spike trains and ANN activations during conversion.

The proposed method is evaluated on several benchmark datasets using ANN-SNN conversion pipelines. Results suggest that TWE achieves lower conversion error and higher accuracy under low timestep budgets compared to traditional encoding methods.

**Compliance With Llm Reviewing Policy:**

Affirmed.

**Final Justification:**

The authors have adequately addressed my concerns around ablation, scalability and timesteps in their rebuttal. As such I have revised my score.

**Key Questions For Authors:**

1. Several prior works explore temporal weighting or multi-spike temporal coding schemes. Could the authors clarify how TWE fundamentally differs from these approaches?

2. How much of the performance improvement comes from the temporal weighted encoding itself versus the proposed temporal and threshold relaxation mechanisms?

3. Have the authors tested TWE on larger networks or datasets (e.g., ImageNet-scale architectures)? Understanding scalability would strengthen the paper’s claims.

4. Since one motivation of ANN-SNN conversion is energy efficiency, can the authors provide estimates or experiments demonstrating the hardware efficiency of TWE-based SNNs?

5. How sensitive is the method to different timestep budgets? A more systematic analysis across different TTT values could help characterize its advantages.

**Limitations:**

Limitations of existing encoding schemes are discussed but potential limitations with TWE could be discussed more fully, such as when the number of timesteps becomes large, possible sensitivity to noise and compatibility with different neuron models and hardware platforms.

**Strengths And Weaknesses:**

Strengths

- The paper addresses an important challenge in ANN–SNN conversion: the trade-off between accuracy and inference latency caused by limited spike timesteps. The observation that common encoding schemes utilize only a small fraction of the theoretical spike sequence space is insightful and clearly illustrated in the paper’s encoding analysis.

-Temporal Weighted Encoding is conceptually straightforward. By assigning exponentially decaying weights to spikes, the scheme effectively transforms spike trains into a form of temporal binary encoding, increasing representational capacity without requiring complex neuron dynamics. This simplicity makes the approach potentially attractive for practical deployment.

-The paper provides a useful conceptual framework for analyzing spike encoding schemes in terms of information capacity and redundancy. The comparison between rate coding, TTFS coding, and the proposed method helps clarify the limitations of existing approaches.

-The empirical results demonstrate that TWE achieves competitive or improved accuracy with fewer timesteps, which is an important metric for real-world SNN deployment where latency and energy efficiency are critical.



Weaknesses

-While the idea of treating spike trains as weighted temporal sequences is interesting, similar concepts have been explored in prior work on temporal coding, weighted spike integration, and binary temporal representations. The paper would benefit from a clearer explanation of how TWE fundamentally differs from these earlier approaches.

-The experiments primarily evaluate classification performance under ANN-SNN conversion. Additional experiments such as evaluations on larger datasets, deeper networks, or event-based tasks would help demonstrate the robustness and generality of the proposed encoding scheme.

-One of the key motivations for SNNs is energy-efficient neuromorphic deployment. However, the paper does not include experiments on neuromorphic hardware platforms.

Ablation studies could be stronger as the experiments do not fully isolate the contribution of each component (TWE, temporal relaxation, threshold relaxation)

-The derivation of the recursive weighting mechanism is not fully intuitive and the description of temporal mismatch and its correction could be expanded.

---

> ### Author Rebuttal · Authors · 2026-03-31
>
> We sincerely thank the reviewer for recognizing the importance of our problem, the conceptual clarity of TWE, the useful encoding analysis framework, and the strong empirical results under low-timestep budgets. We address each concern below.
>
> ---
>
> **Q1: How TWE differs from prior temporal weighting approaches**
>
> We clarify the key distinctions from related work:
>
> - Phase coding: Shares the same power-of-two weight structure but operates offline — waiting for all inputs before computing outputs. Phase coding also uses time-varying thresholds and spike magnitudes. TWE is the first to enable online weighted encoding through relaxation mechanisms that resolve temporal mismatch.
> - FS coding: Similar to phase coding. It uses time-varying thresholds and spike magnitudes. TWE achieves the same capacity with *fixed thresholds and identical spike magnitudes* via recursive integration, avoiding temporal modulation complexity.
> - Burst coding: Allows multiple spikes per timestep, requiring complex burst detection logic. TWE maintains the standard single-spike-per-timestep constraint.
> - Multi-threshold neurons: Encode weights through multiple threshold levels. TWE encodes weights temporally through recursion, requiring only a single threshold.
>
> The core innovation is the recursive integration + relaxation combination: recursive integration provides the simplest mechanism for exponential-capacity encoding, while relaxation mechanisms resolve the inherent temporal mismatch to enable fast online operation.
>
> ---
>
> **Q2: Component contribution breakdown**
>
> We provide an ablation isolating each component's contribution:
>
> | Configuration | ResNet18 on CIFAR10 | VGG16 on CIFAR100 | ResNet34 on ImageNet |
> |--------------|---------------|---------------|---------------|
> | Weighting only (no relaxation) | 23.31% | 3.02% | 0.31% |
> | Weighting + $T_r=1$ | 81.84% | 56.13% | 34.56% |
> | Weighting + Threshold relaxation | 94.04% | 74.55% | 68.09% |
> | TWE (Threshold relaxation + $T_r=1$) |  96.34% | 77.20% | 75.34% |
>
> As shown above, for ResNet18 on CIFAR10, (1) threshold relaxation alone boosts accuracy from 23.31% to 94.04%, demonstrating its critical role. (2) Temporal relaxation provides additional gains, particularly for inputs with highly skewed distributions. (3) The combination achieves the best accuracy-efficiency trade-off.
>
> ---
>
> **Q3: Scalability to larger networks/datasets**
>
> TWE has been extensively evaluated on ImageNet-scale architectures, demonstrating strong scalability:
>
> | Architecture | ANN Acc. | SNN Acc. | $\Delta$ Acc. | T |
> |-------------|----------|----------|--------------|---|
> | ResNet-34   | 75.53%   | 75.51%   | -0.02%       | 3 |
> | VGG-16      | 74.33%   | 74.32%   | -0.01%       | 4 |
> | ViT-S/16    | 81.51%   | 81.59%   | +0.08%       | 5 |
> | ViT-L/16    | 83.81%   | 83.72%   | -0.09%       | 5 |
>
> Near-lossless conversion is maintained across both CNN and Transformer architectures, with ViT-L/16 (300M+ parameters) achieving 83.72% at only 5 timesteps. TWE's encoding capacity is independent of network size, and the bounded residual distribution (Theorem 4.4) ensures stable behavior at any depth.
>
> ---
>
> **Q4: Hardware efficiency estimates**
>
> We acknowledge that our energy analysis is based on operation counts rather than direct hardware measurements. Operation-based estimation is standard practice in the ANN-SNN conversion literature due to the complexity of neuromorphic hardware deployment. TWE's hardware advantages stem from:
>
> - Reduced timesteps: Directly reduces both AC operations and end-to-end latency
> - Zero-cost recursive weighting: The ×2 operation is implemented by simple wiring (connecting low bits of the membrane potential register to high bits of the adder input), contributing no additional energy
> - Signed spike support: Already supported on platforms like Intel Loihi 2 with only a single sign bit overhead per event
>
> With corrected energy values (see response to reviewer j54g), TWE achieves $\approx$51% energy reduction compared to rate coding, confirming the algorithmic advantage translates to hardware efficiency.
>
> ---
>
> **Q5: Sensitivity to timestep budgets**
>
> TWE's exponential encoding capacity ($2^T$) means each additional timestep doubles precision. We evaluate performance across timestep budgets:
>
> | Dataset | T=1 | T=2 | T=3 | T=4 | T=5 | T=6 |
> |---------|-----|-----|-----|-----|-----|-----|
> | VGG16 on CIFAR10 | 89.75% | 95.26% | 95.66% | 95.65% | 95.65% | 95.67% |
> | VGG16 on ImageNet | 0.66% | 66.01% | 73.72% | 74.32% | 74.34% | 74.34% |
>
> TWE achieves strong performance even at minimal timesteps (T=2-3) and scales gracefully. The key advantage over rate coding is that TWE's precision grows exponentially with T, while rate coding grows only linearly.

---

> > ### Author Rebuttal · Reviewer_C6tb · 2026-04-06
> >
> > Thank you or this detailed rebuttal, which addresses all of my main concerns.

---

### Official Review · Reviewer_kmdc · 2026-03-12

**Soundness:** 3
**Presentation:** 3
**Significance:** 2
**Originality:** 2
**Overall Recommendation:** 2
**Confidence:** 3

**Summary:**

This paper proposes Temporal Weighted Encoding (TWE) for ANN-SNN conversion, using recursive integration to realize temporally weighted spike codes and introducing temporal and threshold relaxation to reduce conversion error at low timesteps.

**Compliance With Llm Reviewing Policy:**

Affirmed.

**Key Questions For Authors:**

See above.

**Limitations:**

No, see above.

**Strengths And Weaknesses:**

Strengths:

- The paper studies a relevant problem in ANN-SNN conversion: how to improve encoding fidelity under small timestep budgets without resorting to explicitly complicated neuron dynamics.
- The empirical evaluation is fairly broad, covering CNN and Transformer backbones, static image benchmarks, and several additional tasks beyond image classification. The low-timestep results are strong, especially the 3-step CIFAR and 5-step ImageNet settings.
- The paper provides a clear temporal-mismatch motivation, theoretical discussion of temporal/threshold relaxation, and ablations on both accuracy and estimated energy.

Weaknesses:

- The paper could better isolate its novelty beyond prior phase-coding and signed-spike conversion methods; the main addition seems to be the recursive implementation plus the relaxation analysis.
- The hardware-friendly claim is not fully convincing, since FP32 GPU experiments may not reflect 4-bit or 8-bit neuromorphic deployment where repeated *2 amplification could cause overflow, saturation, or larger quantization error; moreover, the signed-spike correction assumes support for signed event accumulation, whose hardware cost is not analyzed.
- The paper does not report the post-quantization activation sparsity/quantization characteristics or explain why converting the quantized ANN to an SNN is preferable to running the quantized ANN directly.
- The comparisons focus mainly on other conversion methods; direct low-timestep SNN baselines would make the case stronger.

---

> ### Author Rebuttal · Authors · 2026-03-31
>
> We sincerely thank the reviewer for recognizing our problem relevance, evaluation breadth, and strong low-timestep results. We address each concern below.
>
> ---
>
> **Q1: Novelty beyond phase coding and signed-spike methods**
>
> We appreciate this important point. Recursive integration and relaxation mechanisms are not incremental additions, but form a cohesive solution to an open problem in online weighted spike encoding. Recursive integration provides the simplest mechanism to realize exponential-capacity encoding (power-of-two weights) using fixed thresholds and identical spike magnitudes, without time-varying modulation. However, this structure inherently creates temporal mismatch, where small-value dominated inputs accumulate too slowly to trigger early high-weight spikes. Relaxation mechanisms are the principled solution to this problem.
>
> These components are deeply interconnected: without relaxation, recursive integration cannot function reliably online. Prior phase coding works use the same *weight structure* but operate exclusively offline (waiting for all inputs before computing), precisely because they lack a mechanism to resolve this mismatch. TWE is the first to enable fast weighted encoding through relaxation.
>
> In previous literature, negative spikes were introduced to handle negative input currents in the temporal domain, reducing encoding error from bidirectional signals. In TWE, signed spikes serve a completely different purpose: threshold relaxation intentionally lowers the firing threshold to encourage earlier responses, which risks over-firing. Negative spikes are introduced specifically to correct this deliberate overshoot, enabling aggressive yet accurate encoding.
>
> ---
>
> **Q2: Hardware-friendly claim under quantized deployment**
>
> We appreciate this concern and clarify several points:
>
> - Recursive integration does not inherently cause overflow. If we simply accumulate inputs without firing, the total weighted membrane potential is *exactly equivalent* to the offline phase coding result. The ×2 amplification is the mathematical mechanism that produces the correct weights — it does not introduce numerical instability by itself. Under the same membrane potential precision budget, TWE is guaranteed to achieve at least the same encoding precision as phase coding in the worst case.
> - Theorem 4.4 proves that with threshold relaxation, the membrane potential converges to a bounded uniform distribution, which keep the membrane potential within a predictable, hardware-friendly range. Moreover, in neuromorphic hardware, membrane potential precision is typically much higher than weight precision (e.g., 16-32 bit fixed point vs. 4-8 bit weights), which provides ample headroom for the bounded accumulation dynamics.
> - Signed spikes are already supported on neuromorphic platforms (e.g., Intel Loihi 2) and require only an additional sign bit per event.
>
> ---
>
> **Q3: Why convert to SNN rather than directly deploy quantized ANN?**
>
> Quantized ANNs perform **synchronous, dense computation** — every neuron computes at every layer regardless of input. SNNs are **event-driven** — computation occurs only when spikes arrive, enabling massive energy savings in sparse activity regimes. As shown in Table 2, TWE's AC operations (149.83M) are substantially lower than the dense MAC operations of a quantized ANN (313.60M). We also note that converting activation-quantized ANNs to SNNs is the standard and mainstream practice in the SNN community, as it reduces the number of activation levels to be encoded. This approach leverages the well-established ANN training pipelines while enabling event-driven inference on neuromorphic hardware.
>
> ---
>
> **Q4: Missing direct SNN training baselines**
>
> We appreciate this question and would like to clarify a fundamental distinction. Direct SNN training with surrogate gradients does not involve an explicit encoding scheme — it formulates the entire problem as gradient-based optimization, where how neurons encode information remains implicit and uninterpretable. In contrast, ANN-SNN conversion is centered precisely on designing explicit, analyzable encoding schemes that map ANN activations to spike trains. These are fundamentally different research paradigms with different objectives.
>
> Our paper focuses on the conversion paradigm because we aim to understand and optimize the encoding process itself — something that direct training does not explicitly address. That said, we agree that comparing final task performance across paradigms would be informative, and we will add a brief discussion of this in the revision.

---

> > ### Author Rebuttal · Reviewer_kmdc · 2026-04-03
> >
> > Thank you for the rebuttal. It improves the motivation, but I am still not fully convinced.
> >
> > The novelty still seems somewhat limited, as the main contribution appears to be the recursive implementation plus the relaxation analysis, rather than a fundamentally new encoding principle beyond prior phase-coding and signed-spike methods.
> >
> > I am also not convinced by the argument comparing converted SNNs with quantized ANNs. A low-bit or binary quantized ANN may also admit efficient accumulation-based computation rather than simply dense MACs. In particular, with binary activations {0,1}, computation can effectively reduce to conditional accumulation. By contrast, SNNs introduce an additional temporal dimension and may require multiple timesteps to represent the same information. Therefore, comparing SNN AC counts against quantized ANN MAC counts alone is not sufficient to establish that conversion is the preferable deployment choice.
> >
> > Finally, I do not think the absence of direct low-timestep SNN baselines is fully addressed. Although direct training and conversion are different paradigms, such comparisons would still be informative for evaluating practical competitiveness.

---

> > > ### Author Response · Authors · 2026-04-04
> > >
> > > We thank the reviewer for the follow-up feedback. We provide additional clarifications below.
> > >
> > > ---
> > >
> > > Q1: We argue that recursive dynamics and relaxation mechanisms solve important unresolved problems in phase coding and related approaches (phase coding [1][2], FS coding [3], temporal pattern coding [4]). This establishes TWE as a novel encoding scheme. However, if the use of weights alone is sufficient to classify a method as phase coding, then all of the above would also fall into this category. We believe this definition could overlook innovations in core encoding principles.
> > >
> > > First, relaxation mechanisms resolve the intrinsic temporal mismatch of exponentially-weighted encoding, enabling low-latency online operation. **Prior methods did not solve this problem; their papers explicitly identify offline latency as a major limitation (Limitation (2) Sec. V.C [2], Page 2 right col para 3 [3], Sec. IV Discussion para 6 [4]). Resolving this long-standing limitation is a core TWE contribution.**
> > >
> > > Second, recursive implementation is not just an engineering detail, but rather, a core component of the encoding principle. Through recursive dynamics, TWE realizes exponential weight structures without external modulation. **This design fundamentally contrasts with prior phase coding and represents a major shift in weighted spike encoding conception. Explicit weighting complicates hardware design (Sec. 5.4 [1], Page 6 left col [3], Sec. IV Discussion para 7 [4]), an issue TWE avoids entirely.**
> > >
> > > **Regarding signed spikes, such mechanisms have not appeared in prior weighted encoding schemes.** In TWE, negative spikes correct over-firing from threshold relaxation, rather than handling negative input currents as in rate-based methods. Their purpose and mechanism in TWE are fundamentally different from prior uses.
> > >
> > > [1] Deep neural networks with weighted spikes
> > > [2] One-Spike SNN: Single-Spike Phase Coding with Base Manipulation for ANN-to-SNN Conversion Loss Minimization
> > > [3] Optimized spiking neurons can classify images with high accuracy through temporal coding with two spikes
> > > [4] Temporal Pattern Coding in Deep Spiking Neural Networks
> > >
> > > ---
> > >
> > > Q2: We acknowledge QNNs can, in theory, be optimized to use only accumulation operations. However, such designs would operate as time-unfolded SNNs rather than conventional QNNs, and this paradigm is not used in practice for standard QNN inference.
> > >
> > > The core distinction: SNNs are event-triggered — computation occurs only where spikes arrive, with idle neurons consuming near-zero power. QNNs, even with binary activations, evaluate every neuron at every layer on every forward pass, regardless of sparsity. This is exactly why neuromorphic platforms (Loihi, TrueNorth, SpiNNaker) are designed for event-driven SNN computation rather than synchronous quantized ANN computation. **We believe that SNNs' asynchronous architecture and inherent sparsity deliver unique latency/energy advantages justifying the entire SNN research field.**
> > >
> > > We clarify that comparing SNN energy against quantized ANN baselines via operation counts is standard practice across SNN literature, for both conversion and direct training. **This framework is adopted field-wide, including top venue publications (ICML 2025 [1][2], ICLR 2026 [3][4]). Our analysis follows this convention.**
> > >
> > > [1] Differential Coding for Training-Free ANN-to-SNN Conversion
> > > [2] TTFSFormer: A TTFS-based Lossless Conversion of Spiking Transformer
> > > [3] Biologically Plausible Learning via Bidirectional Spike-Based Distillation
> > > [4] Advancing Spatiotemporal Representations in Spiking Neural Networks via Parametric Invertible Transformation
> > >
> > >  ---
> > >
> > > Q3: We agree such comparisons provide useful practical context. While conversion papers typically focus on conversion-specific baselines (the two paradigms address different questions: conversion aims for lossless encoding of pre-trained ANN activations, while direct training learns spike representations end-to-end), we provide a comparison against a strong direct training baseline for completeness:
> > >
> > > | Method | Arch. | Dataset | SNN Acc. | T |
> > > |--------|-------|---------|----------|---|
> > > | Ternary Spike | ResNet-19 | CIFAR-10 | 95.80% | 2 |
> > > | TWE | ResNet-18 | CIFAR-10 | 96.34% | 3 |
> > > | Ternary Spike | ResNet-34 | ImageNet | 70.74% | 4 |
> > > | TWE | ResNet-34 | ImageNet | 75.51% | 3 |
> > >
> > > Ternary Spike [1] is a direct training method that also uses negative spikes, making it a relevant comparison. TWE achieves competitive or superior accuracy even against optimized direct training baselines, while retaining conversion benefits (stable weight inheritance, no training-inference mismatch).
> > >
> > > [1] Ternary Spike: Learning Ternary Spikes for Spiking Neural Networks
> > >
> > > ---
> > >
> > > We sincerely thanks to your kind review again and we believe the above clarifications resolve the main concerns raised in the review, and we would greatly appreciate it if you could take them into account when updating your assessment.

---

### Decision · Program_Chairs · 2026-04-30

**Decision:**

Accept (regular)

**Comment:**

This paper focuses on improving the efficiency and expressiveness of spike encoding in ANN-to-SNN conversion methods under tight timestep budgets. It introduces Temporal Weighted Encoding (TWE), which assigns exponentially decaying weights to spikes via a simple recursive integration, effectively treating them as a temporal code with higher encoding capacity. The authors further identify a temporal mismatch issue arising from this weighting scheme and introduce a temporal relaxation (delayed firing) and threshold relaxation (lowered thresholds with negative spikes) technique to correct it with few timesteps. Empirically the proposed method is shown to perform and scale well.

The paper went through detailed discussions during the rebuttal phase, and reviewers acknowledged the merit of this submission. On the negative side, some reviewers raised concerns about the lack of experiments on neuromorphic hardware platforms, given that energy-efficient neuromorphic deployment is a key motivation for SNNs. The authors provided a detailed qualitative discussion of the hardware compatibility of their approach, arguing that the required modifications to standard IF neuron hardware are minimal and that no fundamental barriers to neuromorphic implementation exist. The AC finds this discussion sufficiently convincing and does not consider the absence of physical hardware experiments a critical shortcoming at this stage. Related discussions and limitations should be included in the paper revisions nevertheless.

There were also concerns regarding the practical preferability of this approach in terms of energy-efficiency. One reviewer challenged the fairness of the efficiency comparison between converted SNNs and quantized ANNs. The authors responded by highlighting the fundamental architectural distinction between event-driven SNN computation and synchronous quantized ANN evaluation. The reviewer also critically noted that the absence of directly trained low-timestep SNN baselines to evaluate practical competitiveness. The authors provided additional experimental comparisons to a relevant direct training baseline that also uses negative spikes, showing that TWE works fine.

Overall, the proposed methodology demonstrates competitive performance while requiring fewer timesteps in ANN-to-SNN conversion, which the AC finds to be an important and well-presented contribution to the area of ANN-to-SNN conversion algorithms. The rebuttal phase appeared successful, with responses that seem to satisfactorily address the open concerns raised by the reviewers with negative scores. This is a conclusion the AC reached upon independent inspection, as these reviewers did not respond during the final justification phase. Combined with two other reviewers endorsing acceptance, the AC also recommends acceptance of this work. The authors are also kindly asked to incorporate the discussions and new results from the rebuttal within their camera-ready manuscripy, and also make their code/implementations publicly available to foster reproducibility.